# Dermo-Cosmetic Benefits of Marine Macroalgae-Derived Phenolic Compounds

Haresh S. Kalasariya [1] and Leonel Pereira [2],*

1    Centre for Natural Products Discovery, School of Pharmacy and Biomolecular Sciences,
     Liverpool John Moores University, Byrom Street, Liverpool L3 3AF, UK
2    MARE—Marine and Environmental Sciences Centre, Department of Life Sciences, University of Coimbra,
     Calçada Martim de Freitas, 3000-456 Coimbra, Portugal
*    Correspondence: leonel.pereira@uc.pt

**Abstract:** Marine macroalgae have an interesting profile of bioactive compounds and have gained tremendous attention in cosmeceuticals with negligible toxicity effects (cytotoxicity, reproductive toxicity, genotoxicity, mutagenicity, carcinogenicity, etc.) on humans and exhibit strong benefits for the skin. Among the diversified compounds, phenolic compounds are the group of phytochemicals found in high amounts with great structural diversity. Phlorotannin is the most studied polyphenol compound in brown algae, but besides there are some other phenolic compounds observed and studied in macroalgae such as terpenoids, bromophenols, mycosporine amino acids (MAAs), and flavonoids. These compounds are already characterized and studied for their full range of cosmeceutical benefits such as skin whitening, moisturizing, photoprotection, antiaging, antiwrinkle, anti-melanogenic, and antioxidant activities as well as in the treatment of pruritus (caused by acne, eczema, dermatitis, hives, psoriasis), photoaging, and skin pigmentation disorders (hypopigmentation due to the absence of melanocytes and hyperpigmentation caused by skin irritation or metabolic disorders). This review study mainly focuses on marine algae-derived phenolic compounds and their extraction, characterization, and skin cosmetic benefits described in the literature. The present study aims to provide a detailed insight into the phenolic compounds in marine algae.

**Keywords:** cosmetics; marine algae; polyphenols; phlorotannin; skin benefits

## 1. Introduction

Cosmeceutical ingredients are active compounds that are used to improve the appearance of the human body and represent a new category of preparations placed between cosmetics and pharmaceuticals. Cosmeceutical formulations intend the improvements of skin health and beauty [1–3]. Globally, the cosmeceutical sector is growing each year due to increasing modern beauty trends. To meet consumer demand, industries are moving towards the excessive use of synthetic cosmetic ingredients in formulations listed as Hydroquinone (HQ), Phthalates, Para-aminobenzoic acid (PABA), Benzophenones, Butylated Hydroxyanisole (BHA), Butylated Hydroxytoluene (BHT), and Dibenzoylmethane (DBM). According to SCCS (Scientific Committee on Consumer Safety) opinion (SCCS/1564/15), the excessive use of synthetic ingredients in cosmeceutical formulations may lead to different types of toxicities such as acute toxicity, corrosion and irritation, skin sensation, dermal/percutaneous absorption, repeated dose toxicity, reproductive toxicity, mutagenicity/genotoxicity, carcinogenicity, and photoinduced toxicity on the skin as well as human health. Hydroxyanisole, widely used in skin-whitening creams, has reported many harmful effects such as ochronosis and potential mutagenicity [4–6]. Benzophenones, DBM, and PABA have shown allergic phototoxicities, dermatitis, and skin irritations [7,8]. Besides, BHA and BHT are applied in moisturization and lipstick preparations that cause allergic reactions, irritation, and corrosivity in the skin. Another ingredient, parabens, are highly carcinogenic and neurotoxic among other harmful health effects. Around 75 to 90 percent

of commercially available products contain parabens, which are mostly used as a mixture in cosmetic formulations. Parabens have been reported to have a high risk of breast cancer and the development of malignant melanoma in women [9]. However, in the ACDS Contact Allergen Management Program (CAMP) report, about 19% of products contained different types of parabens, mainly methylparaben, ethylparaben, propylparaben, and butylparaben. According to them, these components have little allergenicity compared to other preservatives, with no adverse reactions, and low toxicity, safety, and cost [10]. Hafeez and Maibach [11] reported fewer sensitizing effects of parabens in commercial applications but very limited reports are often attributable to the use of parabens on damaged skin. Polyethylene glycol (PEG) is a genotoxic compound that irritates and causes systemic toxicities and skin damage. In skin cosmetics, PEGs function in three ways: as emollients (that soften and lubricate the skin), as emulsifiers (that help to mix water-based and oil-based ingredients properly), and as vehicles (that deliver ingredients deeper into the skin). In addition, the Agency for Toxic Substances and Disease Registry (ATSDR) Information Center, and the Centers for Disease Control and Prevention (CDC) reported the toxicity of dibutyl phthalate (DBP) in DNA damage in male reproductive cells [12]. Some previous studies have reported the harmful effects of cosmetic ingredients in animal studies, such as male genitalia disabilities that altered pregnancy outcomes as well as reduced sperm counts [13,14]. Moreover, the EC 1223/2009 regulation regarding the testing and marketing ban of cosmetic products suggests the prohibition of testing finished cosmetic products on animals and their marketing.

To overcome the toxicities of these formulations, consumers have changed their preference to natural skin care products in the last few years. As a result, industries have moved towards natural bioactive ingredients from various natural resources that are eco-friendly and less toxic [15,16]. Various natural resources can be used in skin cosmetic products such as terrestrial plants, fungi, marine algae, bacteria, animals, etc. [17–21]. Among them, marine macroalgae are widely utilized for their skin benefits nowadays. Marine macroalgae are also known as seaweed: eukaryotic, aquatic photosynthetic macroscopic, multicellular organisms that are ubiquitously found along the seacoast and in seawater. They belong to the Eukaryota domain and are classified into three major taxonomic groups, red algae, brown algae, and green algae, belonging to the Rhodophyta phylum, Ochrophyta phylum, and Chlorophyta phylum, respectively [22–25]. These different types of marine macroalgae are illustrated in Figure 1. There is an increasing demand for bioactive constituents in cosmetic and cosmeceutical applications from macroalgae. The applications of macroalgae-derived compounds to the cosmetic industry are based on their potential biological activities [26–28]. These are lipids, fatty acids, polysaccharides, vitamins, minerals, amino acids, phenolic compounds, proteins, pigments, etc., which have attracted attention for their skin cosmeceutical benefits [29–31].

Marine algae are one of the natural resources of phytochemical compounds, which confer potential biological activities [32,33]. Phenolic compounds are one of the bioactive compounds produced in seaweeds, are made of an aromatic ring with one or more hydroxyl groups, and their structures diversify from simple to complex, higher molecular weight compounds [34,35]. Many previous studies have been carried out in which phenolic compounds were isolated from marine algae and they include simple phenolic compounds or polyphenols such as flavonoids, phlorotannins, mycosporine-like amino acids (MAAs), bromophenols, and terpenoids [36]. The biological action of phenolic compounds is determined by the position of the hydroxyl groups, and the number of phenyl rings in the structure [37]. Brown algal species contain a high amount of phlorotannins whereas green and red algae mainly produce flavonoids, bromophenols, terpenoids, and mycosporine amino acids in response to environmental conditions [36–40]. Marine algae-derived phenolic compounds have a wide variety of applications such as enzyme inhibitory effects (for example, tyrosinase inhibition, elastase inhibition, collagenase inhibition, matrix metalloproteinase inhibition in photoprotection, inhibition of angiotensin-converting enzyme-1 (ACE-1), pro-inflammatory cyclooxygenase and lipoxygenase (COX-1, 2 and 5-LOX) as well

as dipeptidyl peptidase-4 (DPP-4) inhibition, and hydroxymethyl glutaryl coenzyme A reductase (hMGCR) inhibition) antibacterial, antifungal, antioxidant, and anti-inflammatory properties, which can be very attractive when utilized in cosmetics and cosmeceutical product preparations [41–47]. In cosmetics, phlorotannin provides hyaluronidase activation, antiallergic, anti-wrinkle, anti-aging, skin whitening, photoprotection, and skin health improvement benefits [48,49]. This review represents marine algae-derived phenolic compounds, their chemical structures, together with their skin benefits and their potential in the skin cosmetics industries.

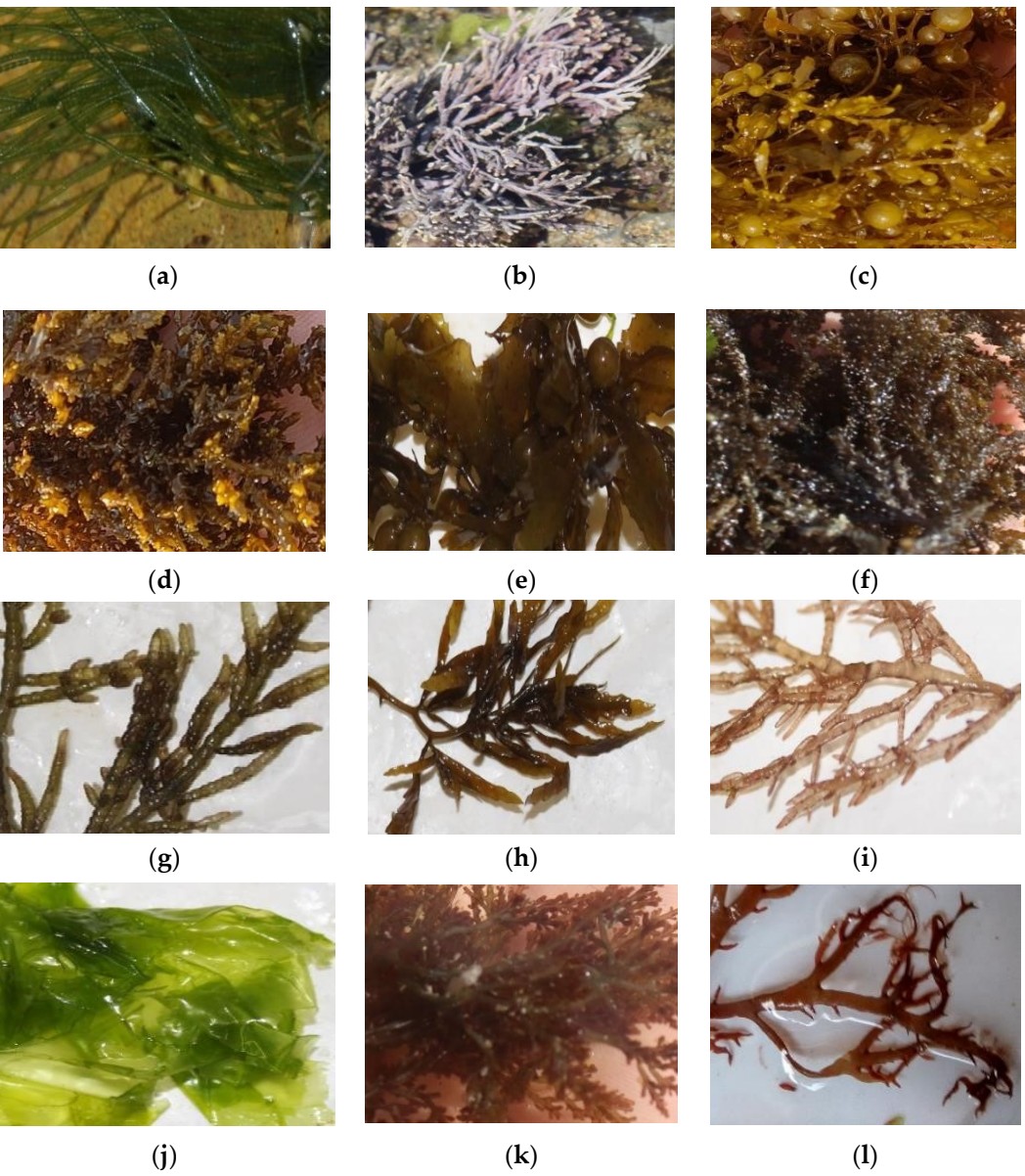

**Figure 1.** *Cont*.

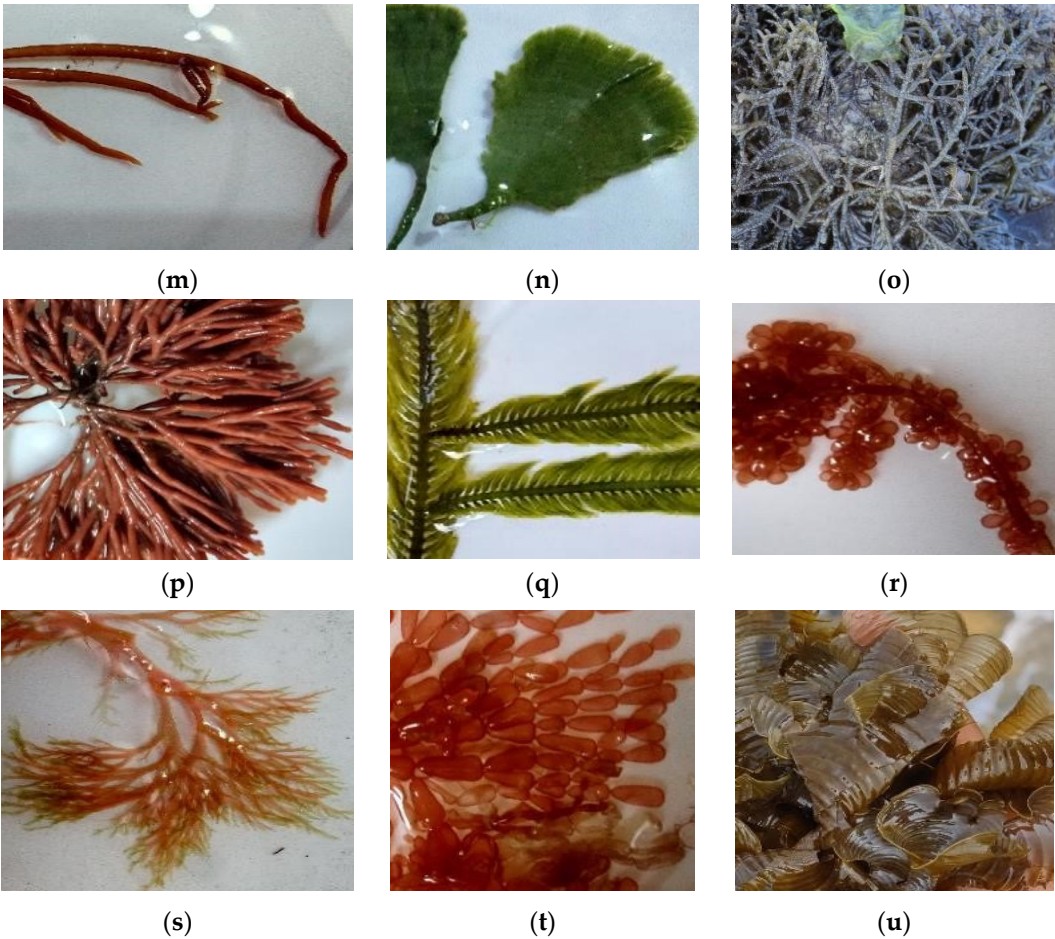

**Figure 1.** (**a**–**u**) Examples of marine macroalgae. (**a**) *Chaetomorpha antennina* (green); (**b**) *Corallina* sp. (red); (**c**) *Sargassum linearifolium* (brown); (**d**) *Palisada perforata* (red); (**e**) *Sargassum cinereum*; (**f**) *Palisada perforata* (as *Laurencia papillosa*); (**g**) *Laurencia glandulifera* (red); (**h**) *Sargassum tenerrimum* (brown); (**i**) *Sargassum tenerrimum* (brown); (**j**) *Ulva lactuca* (green); (**k**) *Laurencia* sp. (red); (**l**) *Gracilaria* sp. (red); (**m**) *Gracilaria debilis* (red); (**n**) *Udotea indica* (green); (**o**) *Champia compressa* (red); (**p**) *Tricleocarpa fragilis* (red); (**q**) *Caulerpa taxifolia* (green); (**r**) *Botryocladia leptopoda* (red); (**s**) *Centroceras clavulatum* (red); (**t**) *Scinaia moniliformis* (red); (**u**) *Padina tetrastromatica* (brown).

Phenolic compounds are one of the most researched marine macroalgae-derived biologically active compounds and are already utilized in various cosmeceutical preparations [50]. Normally, these phenolic compounds are not extracted because seaweed extracts contain a good number of phenolic compounds. According to Thomas and Kim [51], Nagayama et al. [52], and Hwang [53], phlorotannin is one of the marine algae-derived phenolic compounds with less toxicity than other natural antioxidant molecules and its anti-aging benefit is attracting the attention of researchers looking to use it as an ingredient in cosmetic formulations. Some marine algae extracts are rich in different phenolic compounds such as phlorotannin, phloroglucinol, eckol, dieckol, fucol, phlorethol, fuhalols, lignans, bromophenol, flavonoid, phenolic terpenoids, and mycosporine amino acids (MAAs). These phenolic compounds contribute to skin benefits, with antiaging, photoprotection, antiwrinkle, antiallergic, anti-inflammation, antioxidant, antimicrobial, antifungal, tyrosinase inhibition, anti-melanogenic, skin whitening, UVB protector, and antiacne properties, etc. [54–56]. Tang et al. [57], and Khanavi et al. [58] reported not only cytotoxicity but also the antibacterial effect of a phenolic fraction from the *Ulva clathrata* and *Ulva flexuosa* species that can be utilized for skin benefits. In addition, Lavoie et al. [59] identified the antibacterial activity of *Cladophora socialis*-derived phenolic compounds, such as 2,3,8,9-tetrahydroxybenzo[c]chromen-6-one, 3,4,30,40-tetrahydroxy-1,10-biphenyl, and

cladophorol against methicillin-resistant Staphylococcus aureus. Based on a study carried out by Ko et al. [60], bromophenols, such as 5′-Hydroxyisoavrainvilleol found in green macroalga *Avrainvillea nigricans*, demonstrate promising antimicrobial activity. The green macroalga, *Caulerpa* sp., has been reported for various flavonoids, such as kaempferol and quercetin. These compounds have been studied and identified with antioxidant benefits [61]. In addition, *Vidalia colensoi* (as *Osmundaria colensoi*) derived from lanosol methyl ether, lanosol butenone, and rhodomelol has revealed antibacterial and antifungal activity against various bacterial and fungal pathogens. These effects proved to be bactericidal and bacteriostatic or fungicidal, fungistatic, and antiacne, and demonstrated a dose-dependent curve effect against the pathogenic organisms [62]. Mycosporine amino acids such as palythine, shinorine, asterina-330, *Porphyra*-334, palythinol, and usujirene have already been isolated and have antioxidant, photoprotective, and antiproliferative activity in the HeLa and HaCat cancer cell lines. In other studies, Lawrence et al. [63], Orfanoudaki et al. [64], and Becker et al. [65] reported the anti-inflammatory and immune-modulatory properties of mycosporine-like amino acids. These compounds can act as UV filters against photodamage. Moreover, *Ecklonia cava*-derived phlorotannin acts as an anti-UVB protective and reduces the photodamage effect provoked by UVB radiation [66]. Some other brown macroalgal species-derived phlorotannins such as dieckol, dioxinodehydroeckol, eckol, eckstolonol, phlorofucofuroeckol A, and 7-phloroeckol are being researched for skin-whitening and antiwrinkle properties, as well as tyrosinase and hyaluronidase inhibition [67–73]. Bak et al. [74] reported the hair growth-promoting activity of 7-phloroeckol isolated from *E. cava*.

## 2. Characterization and Types of Phenolic Compounds

The extraction and characterization of phenolic compounds from marine algae constitute interesting results, with those reported in the literature [75–79]. These compounds and their biological action are commonly correlated. However, some phenolic extracts have an interesting property but have not been fully characterized. Antioxidant activities have been reported in green seaweed-derived bromophenols and flavonoids. Farasat et al. [80], and Cho et al. [81] studied and proved the high radical scavenging activities of various green (Chlorophyta) species such as *Ulva clathrata*, *U. compressa* (formerly known as *Enteromorpha compressa*), *U. intestinalis*, *U. linza*, *U. flexuosa*, *U. australis* (formerly known as *Ulva pertusa*), *Capsosiphon fulvescens*, and *Chaetomorpha moniligera*. In their findings, antibacterial and cytotoxic effects on breast ductal carcinoma cell lines were verified in the phenolic fraction of *U. clathrata* and *U. flexuosa* [82,83]. In a more recent study, Lavoie et al. [84] reported *C. socialis*-derived phenolic compounds, such as 2,3,8,9-tetrahydrobenzo[c]chromen-6-one (Figure 2a), 3,4,3′,4′-tetrahydroxy-1,1′-biphenyl (Figure 2b), and cladophorol C (α-hydro-ω-[3,4-dihydroxyphenyl]octa[oxy(2-hydroxyphen-4-yl)]) (Figure 2c), have been identified with antibacterial activity against methicillin-resistant *Staphylococcus aureus* (MRSA).

The functions of phenol compounds in red marine algae have barely been studied, but they probably have multipurpose actions in cell life, such as antioxidant, chelation, and anti-infection actions, as well as cofactors or hormones [85]. However, some research is not with the isolated phenolic compounds but with an extract enriched in polyphenolics [85]. More than 8000 different structures of phenolic compounds are found in marine macroalgae, because of their importance in organisms' growth, survival, and defense. The classification of phenolic compounds according to their chemical structures is depicted in Figure 3.

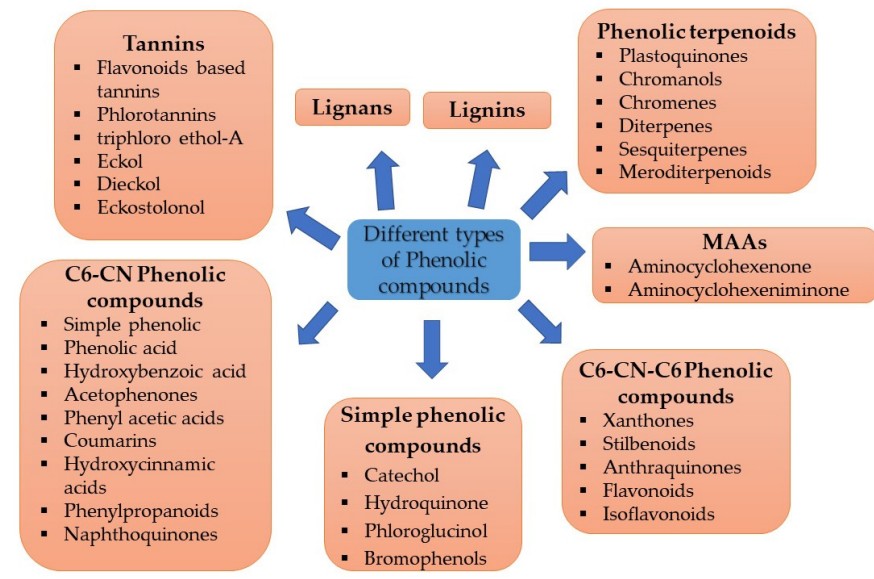

**Figure 2.** (**a**–**c**). Chemical structures of *Cladophora socialis* (green)-derived phenolic compounds (**a**): 2,3,8,9-tetrahydroxybenzo[c]chromen-6-one; (**b**): 3,4,3′,4′-Tetrahydroxy-1,1′-biphenyl; (**c**): New phenolic natural compound. Source (permission obtained from the authors): Lavoie et al. [84].

**Tannins**
- Flavonoids based tannins
- Phlorotannins
- triphloro ethol-A
- Eckol
- Dieckol
- Eckostolonol

**Lignans**   **Lignins**

**Phenolic terpenoids**
- Plastoquinones
- Chromanols
- Chromenes
- Diterpenes
- Sesquiterpenes
- Meroditerpenoids

**Different types of Phenolic compounds**

**C6-CN Phenolic compounds**
- Simple phenolic
- Phenolic acid
- Hydroxybenzoic acid
- Acetophenones
- Phenyl acetic acids
- Coumarins
- Hydroxycinnamic acids
- Phenylpropanoids
- Naphthoquinones

**MAAs**
- Aminocyclohexenone
- Aminocyclohexeniminone

**Simple phenolic compounds**
- Catechol
- Hydroquinone
- Phloroglucinol
- Bromophenols

**C6-CN-C6 Phenolic compounds**
- Xanthones
- Stilbenoids
- Anthraquinones
- Flavonoids
- Isoflavonoids

**Figure 3.** Classification of phenolic compounds listed according to their chemical structures. Source: Cotas et al. [36].

Moreover, the chemical structures of some phenolic compounds are illustrated in Figure 4. These compounds can be synthesized by various metabolic pathways such as the pentose phosphate pathway (PPP), and the phenylpropanoid and shikimate pathways.

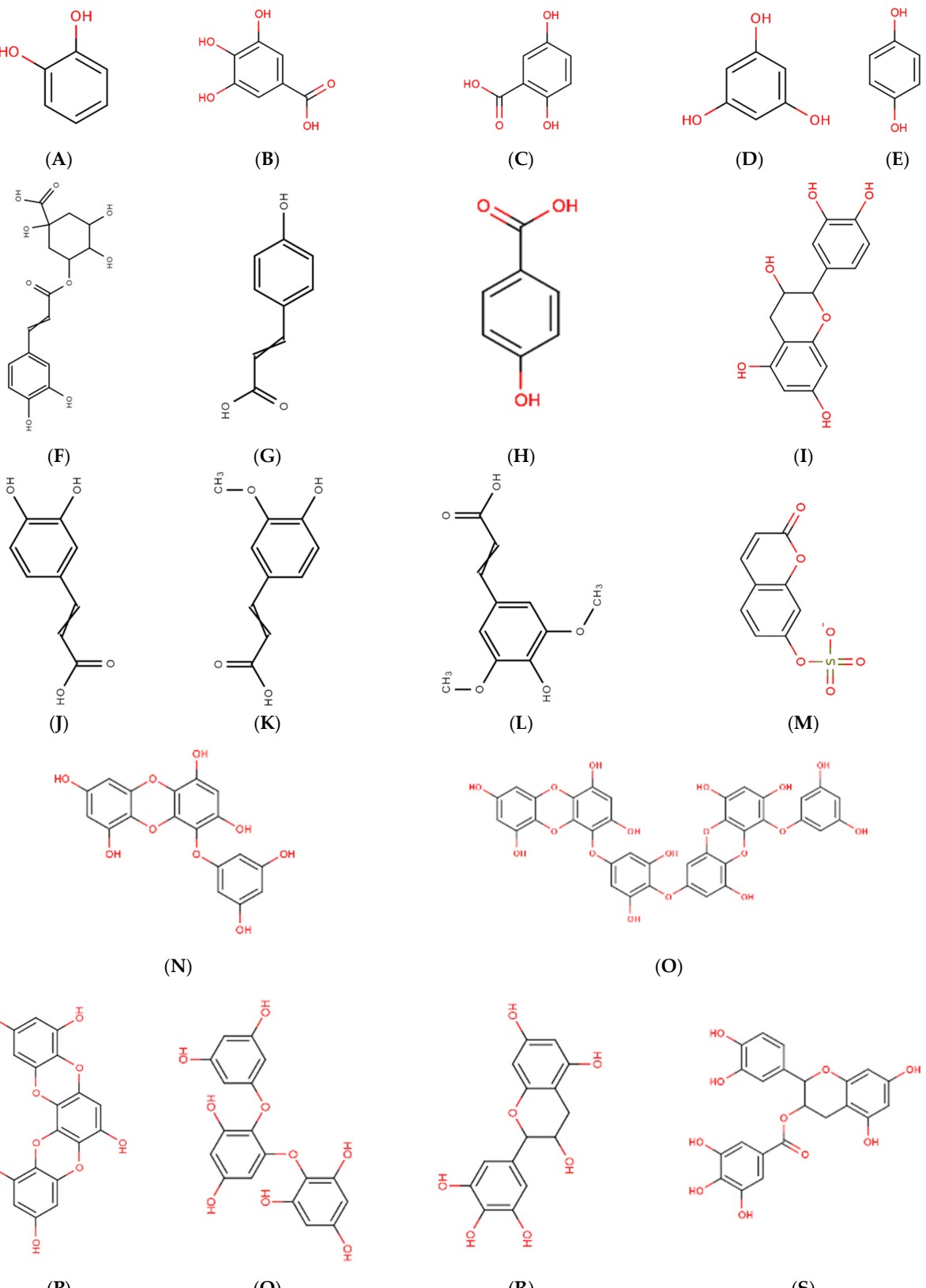

**Figure 4.** Chemical structures of some phenolic compounds. (**A**): Catechol, (**B**): Gallic acid, (**C**): Gentisic acid, (**D**): Phloroglucinol, (**E**): Hydroquinone (HQ), (**F**): Chlorogenic acid, (**G**): Coumaric acid, (**H**): 4-Hydroxybenzoic acid, (**I**): (+)-Catechin, (**J**): Caffeic acid, (**K**): Ferulic acid, (**L**): Sinapic acid, (**M**): 7-Hydroxy coumarin sulfate, (**N**): Eckol, (**O**): Dieckol, (**P**): Eckstolonol, (**Q**): Triphloroethol-A, (**R**): Epigallocatechin, (**S**): Catechin 3-O-gallate.

Giada [86], and Vermerris and Nicholson [87] reported varieties of phenolic compound classification to comprise a large number of heterogeneous structures from simple to highly polymerized structures. A simple phenolic group is formed that reveals hydroxyl groups at different positions: ortho, meta, and para (1,2-, 2,3-, and 1,4-), respectively. Catechol, HQ, and phloroglucinol are examples of simple phenolic compounds found exclusively in macroalgae [88]. One of the previous studies showed catechol in twenty-seven Japanese green and red seaweeds [89]. It is quite common to find this simple phenol with a bromine substituent that is bromophenol. C6-CN phenolic compounds possess a basic C6-CN structure, where N is found between 1 and 3. Within this phenolic group, three subdivisions can be made in C6-C1, C6-C2, and C6-C3 and correspond to phenolic acids and aldehydes, which are characterized by having phenol with carboxylic group substitution [87]. Other examples are phenolic acid, hydroxybenzoic acids (C6-C1), acetophenones, phenylacetic acids, coumarins (C6-C2), coumarins, hydroxycinnamic acids, phenylpropanoids (C6-C3), naphthoquinones (C6-C4), xanthones (C6-C1-C6), stilbenoids, anthraquinones (C6-C2-C6), flavonoids, isoflavonoids (C6-C3-C6), lignans, neolignans ([C6-C3]2), lignins ([C6-C3]n), and condensed tannins ([C6-C3-C6]n). Gallic acid is one of the simplest phenolic acids used as a standard for total phenol content estimation. It can be found in high concentrations in the brown alga *Halopteris scoparia*. Other simple acids such as 4-hydroxybenzoic acid have been also reported in the brown alga *Undaria pinnatifida* [90]. C6-C2 are not very common in nature but the red alga *Tichocarpus crinitus* was studied for the C6-C3 category of phenolic compounds such as coumarins, isocoumarins, chromones, monolignols, hydroxycinnamic acids, and cinnamic aldehydes [87,91]. Moreover, Hartmann et al. [92] found the presence of coumarins in the green algae *Dasycladus vermicularis*. Other phenolic compounds include xanthonoids (C6-C1-C6), stilbenoids, anthraquinones, anthrones (C6-C2-C6), flavonoids (C6-C3-C6), and diarylheptanoids (C6-C7-C6). (C6-C3-C6) can be classified based on the arrangement of the C3 group that connects two benzene rings such as chalcones, aurones, and flavonoids. The last compound is further classified into different classes such as flavonols, flavones, isoflavones, anthocyanins, and flavanones. Cho et al. [81] found a high content of flavonoids in red algae, which was higher than in green and brown algae. In addition, Generalić Mekinić et al. [90] reported a good number of different flavonoids, catechin, epicatechin, gallate, and epigallocatechin, in brown algae species such as *Eisenia bicyclis*, *Sargassum fusiforme*, and *Saccharina japonica*.

### 2.1. Polyphenolic Compounds

Polyphenol is mainly of two types, phlorotannin, and phloroglucinol. The former is a polymer of phloroglucinol with an additional halogen or hydroxyl group whereas the latter contains an aromatic ring structure with three hydroxyl groups [93–95]. These can be subclassified into six different groups: (i) Eckols; (ii) Fucophlorethols; (iii) Fucols; (iv) Phlorethols; (v) Carmalols; and (vi) Fuhalols.

### 2.2. Lignans

Lignans are a type of phenolic compound, a dimer or oligomer, formed due to the union of monolignols, coniferyl alcohol, and sinapyl alcohol. Freile-Pelegrín and Robledo [96] reported the presence of lignans in calcified red marine algae *Calliarthron cheilosporioides* (Rhodophyta). Another polymeric phenol, lignin, is the most abundant organic polymer found in nature but not extensively studied in marine algae, which are structurally composed of monolignols (coniferyl alcohol, sinapyl alcohol), and lignan units randomly linked forming a polymeric network. Tannins are usually divided into three different chemical structures: hydrolyzable tannins, flavonoid-based tannins, and phlorotannins. The first one is derived from simple phenolic acids and their carbohydrate hydroxyl groups that are partially or completely esterified with phenolic groups. The second, flavonoid-based tannins, synthesize through flavins and catechins whereas the last, phlorotannins, are oligomers of phloroglucinol that are exclusively found in brown algae [97].

### 2.3. Phlorotannins

Phlorotannins, a group of compounds that majorly include dioxinodehydroeckol (eckostolonol), dieckol, eckol, phlorofucofuroeckol A, 7-phloroeckol, and fucofuroeckol A and 8,8′-bieckol, exhibit antioxidant-inhibitory effects against melanin synthesis, skin whitening (tyrosinase inhibition), and UV protection [98–103]. Kong et al. [104], Kim et al. [105], Ahn et al. [106], Lee et al. [107], and Li et al. [108] demonstrated the anti-proliferative, anti-inflammatory, and anti-adipogenic activities of *Ecklonia cava* (Phaeophyceae)-derived dioxinodehydroeckol, dieckol, and phlorofucofuroeckol.

Phlorotannins are the most deeply studied phenolic compounds from algae [109]. Their antioxidant power is 2 to 10 times higher when compared to ascorbic acid or tocopherol [110,111], which demonstrates their role as an anti-inflammatory agent [112]. They can act as an anti-UCB protector; Ryu et al. [113] suggested UVB protection by dioxinodehydroeckol from *E. cava* on the HaCat cells that reduce the provoked apoptosis. Moreover, phlorotannins such as dieckol, dioxinodehydroeckol, eckol, eckstolonol, phlorofucofuroeckol A, and 7-phloroeckol isolated from different marine algae are being researched in cosmetics as whiteners and antiwrinkle agents. They have been shown as promising tyrosinase inhibitors and hyaluronidase inhibitors [114–120]. In addition, Bak et al. [121] proved the hair growth-promoting activity of 7- *E. cava*-derived phloroeckol. Several reports have evaluated the effective antibacterial effect of phlorotannins, including from *Ecklonia cava subsp. kurome* (formerly *Ecklonia kurome*) (Phaeophyceae), against several food-borne pathogenic bacteria (e.g., methicillin-resistant *Staphylococcus aureus* (MRSA) strains, *Campylobacter* sp., and *Streptococcus pyogenes*) [122–124].

### 2.4. Bromophenols

Phenolic compounds such as bromophenol and benzoic acids have been fully isolated and characterized from red seaweeds [125]. Pérez et al. [126], Duan et al. [127], and Choi et al. [128] studied the antioxidant activity of *Vertebrata constricta* (formerly *Polysiphonia stricta* or *P. urceolata*) (Rhodophyta)-derived phenolic compounds, but that depends on the brominated units and degree of bromination. In the same study, *Symphyocladia latiuscula*-derived bromophenols reported antioxidant activity that was studied by DPPH assay. Colon et al. [129] found a cytotoxic effect on KB cells (human epithelial carcinoma cells) and the antimicrobial activity of *Avrainvillea nigricans* (Chlorophyta)-derived 5′-hydroxyisoavrainvilleol, which is an example of bromophenol. Moreover, Carte et al. [130] studied rawsonol, an example of bromophenol, isolated from the same genus, but from another species, *A. rawsoni*, which revealed an inhibitory effect in HMG-CoA reductase (a rate-controlling enzyme that produces cholesterol) activity. Besides, Estrada et al. [131] reported the antibacterial activity of brominated monoterpenoid quinol isolated from *Cymopolia barbata* (Chlorophyta) against *S. aureus* and *Pseudomonas aeruginosa*.

### 2.5. Flavonoids

Other classes of phenolic compounds have been investigated for varieties of applications in cosmetics. Tanna et al. [132] found the antioxidant activity of various flavonoids such as kaempferol and quercetin from *Caleurpa* spp. (Chlorophyta). *Acanthophora spicifera* (Rhodophyta)-derived flavonoid demonstrates a mixture of chlorogenic acid (69.64%), caffeic acid (12.86%), vitexin-rhamnose (12.35%), quercetin (1.41%), and catechol (0.59%), and this flavonoid-enriched extract has revealed antioxidant activity [133,134]. These molecules are multi-active components that play a role in UV radiation absorption, the neutralization of ROSs, and the inhibition of radical reactions, etc., which makes them important contributors to cosmeceuticals [135]. This antioxidant activity becomes helpful to overcome photo-induced skin aging. Ultraviolet light produces reactive oxygen species (ROS) in cells that initiate the intracellular and extracellular oxidative stresses that are responsible for wrinkle formation and atypical pigmentation on the skin [136].

### 2.6. Phenolic Terpenoids

Makkar and Chakraborty [137] studied a chromene-based phenolic compound from Gracilaria opuntia (Rhodophyta) that has been reported to have antioxidant activity in in vitro assays. Pillai et al. [138] reported the role of antioxidants in the prevention of extracellular matrix damage, the activation of MMPs, and inhibition of their expression. These molecules scavenge and quench radical oxygen species (ROS). Freile-Pelegrín and Robledo [96] found diterpenes and sesquiterpenes more commonly in red macroalgae as well as in Sargassaceae and Rhodomelaceae. J. Chappell and R. M. Coates [139] showed the role of sesquiterpene patchoulol as an extremely popular fragrance agent in colognes and perfumes. Ruberto and Baratta [140] found the significant lipid oxidation efficacy of oxygenated sesquiterpenes, which contributes to its role as an antioxidant that may act as a eustressor.

### 2.7. Mycosporine-like Amino Acid

Various marine algal species such as *Asparagopsis armata*, *Chondrus crispus*, *Mastocarpus stellatus*, *Palmaria palmata*, *Gelidium* sp., *Pyropia* sp. (formerly known as *Porphyra* sp.), *Gracilaria cornea, Solieria chordalis, Grateloupia lanceola*, and *Curdiea racovitzae* (Rhodophyta) have been investigated for this exclusive class of phenolic compounds. This class of compounds is more commonly found free in the intracellular space and around cell organelles sensitive to ultraviolet rays. Mycosporine-like amino acids (MAAs) are formed by cyclohexenone or cycloheximide chromophore conjugated to imino alcohol or an amino acid residue [141,142]. Various MAAs (palythine, shinorine, asterina-330, Porphyra-334, palythinol, and usujirene) have already been studied that have high antioxidant, photoprotection, and anti-proliferative (HeLa cancer cell line, human cervical adenocarcinoma cell line) and HaCat (human immortalized keratinocyte) activity [143,144]. Recent studies have reported other important activities such as anti-inflammatory, and photoprotective activities (an alternative to the synthetic UV-R filters in sunscreens). Thus, MAAs seems to be a special focus on a specific area and application that can be applied to humans. Based on the literature, the different types of marine algae-derived phenolic compounds and their skin cosmetic benefits are tabulated in the below Table 1.

**Table 1.** Applications of marine macroalgae-derived phenolic compounds in skin benefits.

| No. | Name of Marine Algae | Seaweed-Based Bioactive Compounds | Cosmetic Properties/Benefits | References |
|---|---|---|---|---|
| 1 | Macroalgal species | Catechins, Flavanols, Flavanol glycosides, Gallic acid, Epicatechin, Phloroglucinol, Pyro catechol, Gallate, Flavonoids, Anthocyanins, Stilbenes, Lignans | Matrix Metalloproteinase (MMP) inhibitors, Reduce collagen degradation | [145–147] |
| 2 | *Corallina pilulifera* (R) | - | Inhibit the expression of MMP2 and MMP-9 | [148] |
| 3 | *Sargassum horneri* (B) | Sargachromanol E | Antiaging | [149] |
| 4 | *Phycocalidia vietnamensis* (R) | Mycosporine-like amino acids (MAAs) | UV absorber | [150] |
| 5 | *Ecklonia cava* (B) | Phlorotannins | Skin whitener, Tyrosinase inhibition | [151] |
| 6 | Macroalgal species | - | Antioxidant activity | [152,153] |
| 7 | Macroalgal species | Phlorotannins | Anti-wrinkle, Antiaging | [154,155] |
| 8 | *Sargassum fusiforme* (as *Hizikia fusiformis*) (B) | Phlorotannins | Tyrosinase inhibition, Skin whitener | [156] |
| 9 | *Corallina pilulifera* (R) | Phlorotannins, Eckol, Fucols, Fucophorethols, Fuhalols, Phlorethols | Antiaging, Antiphotoaging, Antioxidant, Tyrosinase inhibition | [157–160] |
| 10 | Macroalgal species | Phlorotannins | Inhibit melanin synthesis, Anti UVB photodamage | [161] |
| 11 | *Ecklonia cava* (B) | Phlorotannins | Melanin synthesis, UV protector | [162,163] |
| 12 | *Ecklonia cava* (B) | Phlorotannins such as eckstolonol, dieckol | Antioxidant, photoprotective, UV protector | [164] |

**Table 1.** *Cont.*

| No. | Name of Marine Algae | Seaweed-Based Bioactive Compounds | Cosmetic Properties/Benefits | References |
|---|---|---|---|---|
| 13 | Brown algae species | Phlorotannins such as Phloroeckol, Tetrameric phloroglucinol | Anti-skin aging, Antioxidant | [165] |
| 14 | *Corallina pilulifera* (R) | Phlorotannins | Metalloproteinase inhibitors, and UV protectors, Prevent collagen degradation, Wrinkle formation | [166] |
| 15 | *Ulva clathrata, Ulva compressa* (as *Enteromorpha compressa*), *Ulva intestinalis, Ulva linza, Ulva flexuosa, Ulva australis, Capsosiphon fulvescens, Chaetomorpha moniligera* (G) | Bromophenols, Flavonoids | Highly radical scavenger | [167,168] |
| 16 | *E. cava* (B) | Phlorotannins | UVB protector | [169] |
| 17 | *Saccharina japonica* (as *Laminaria japonica*), *Ecklonia cava*, (B) | Phlorotannins | Utilized in facial masks, UV protectors, Anti-acne | [170–172] |
| 18 | *Ulva compressa* (as *Enteromorpha compressa*) (G) | Flavonoids, Tannins, Phlorotannins | Antioxidant effect, Anti-aging | [173] |
| 19 | *Fucus vesiculosus* (B) | Flavonoids, Phenols, HQ, Saponin | Tyrosinase inhibitor, Melanin Inhibition | [174] |
| 20 | *Ecklonia cava* (B) | Phlorotannins; Eckol, Dieckol, Dioxinodehydroeckol, 7-phloroeckol, Phloroglucinol | Tyrosinase inhibition (Skin whitener) | [175–177] |
| 21 | *Ericaria selaginoides* (as *Cystoseira tamariscifolia*), *Gongolaria usneoides* (as *Cystoseira usneoides*), *Fucus spiralis*, *Gongolaria nodicaulis* (as *Cystoseira nodicaulis* (B) | Phlorotannins, Fucophloroethol, Bieckol, Phlorofucofuroeckol, 7-phloroeckol | Antioxidant, Anti-aging, anti-wrinkling, Hyaluronidase inhibition, Lipid peroxidase inhibition | [178] |
| 22 | *Ecklonia bicyclis* (as *Eisenia bicyclis*) (B) | Phlorotannins (Phlorofucofuroeckol-A, Dieckol, Eckol, Phloroglucinol, 8,8′ bieckol | Hyaluronidase inhibitor, Anti-wrinkle | [179] |
| 23 | *Ecklonia kurome* (B) | Phlorofucofuroeckol A, 8-8 bieckol, Dieckol, Eckol, Phloroglucinol | Hyaluronidase inhibition, Anti-wrinkle | [180] |
| 24 | *Ecklonia stolonifera* (B) | Phlorotannins: Eckol, Phlorofucofuroeckol A, Dieckol, Eckstolonol | Tyrosinase inhibitor, Skin whitener Metalloproteinase inhibitors, Anti-wrinkle | [181] |
| 25 | *Ecklonia stolonifera* (B) | Phlorotannins: phlorofucofuroeckol A | Anti-inflammatory | [182] |
| 26 | *Ecklonia cava* (B) | Phlorotannins | UVB protector | [183] |
| 27 | *Ecklonia cava* (B) | Phlorotannins, 6,6′-Bieckol, dioxinodehydroeckol | Metalloproteinase inhibitors, Anti-wrinkle | [183] |
| 28 | *Fucus vesiculosus, Ecklonia cava, Corallina pilulifera* (R) | Eckols, Fucols, Fuhalols, Phlorethols, Fucolphlorethols | Antiphotoaging, Antiaging, Antioxidants, UV protector, Tyrosinase inhibition, Hyaluronidase inhibition | [184–187] |
| 29 | *Ishige foliacea* (B) | Octaphlorethol A | Tyrosinase inhibitor (whitening effect) | [188] |
| 30 | *Ishige okamurae* (B) | Diphlorethohydroxycarmalol | Antioxidant, UV protector | [189] |
| 31 | *Sargassum horneri* (B) | Sargachromanol E | Antiaging, Metalloproteinase inhibitors | [189] |
| 32 | *Gracilaria gracilis* (R) | Phenol | Antioxidant, ROS scavenger | [190] |
| 33 | *Sargassum polycystum* (B) | Flavonoids, Tannins, Terpenoids, Phenols, Saponins | Anti-melanogenesis (skin whitener) | [190,191] |
| 34 | *Laurencia* sp. ® | Bromophenols | Antioxidant | [192] |
| 35 | *Halidrys siliquosa, Ecklonia cava, Ascoseira mirabilis, Cystosphaera jacquinotii, Ishige okamurae*, (B) | Phlorotannins: diphlorethol, triphloroethol, trifuhalol and tetrafuhalol, phloroglucinol, eckol, eckstolonol | Antioxidant, UV protector | [193–198] |
| 36 | *Fucus vesiculosus* (B) | high polyphenol content | Increased brightness and skin age spot reduction, UV protector, and soothing benefit | [198] |
| 37 | *Sargassum polycystum, Ecklonia cava* subsp. *stolonifera* (as *Ecklonia stolonifera*), *Ecklonia cava, Sargassum siliquastrum* (B) | Unspecified flavonoids, Tannins, Phlorotannins | Tyrosinase inhibition, Anti melanogenesis | [199–201] |

**Table 1.** *Cont.*

| No. | Name of Marine Algae | Seaweed-Based Bioactive Compounds | Cosmetic Properties/Benefits | References |
|---|---|---|---|---|
| 38 | *Eisenia bicyclis, Ecklonia cava* subsp. *kurome* (as *Ecklonia kurome*), *Ecklonia cava* (B) | Phlorofucofuroeckol-A, Phlorotannins | Hyaluronidase inhibition Anti-inflammatory Inhibit melanin synthesis, Antioxidant | [202] |
| 39 | *Ecklonia stolonifera* (B) | Phlorofucofuroeckol A and B | Anti-inflammation, Antiaging (Metalloproteinase inhibitors) | [203] |
| 40 | *Sargassum fusiforme* (as *Hizikia fusiformis*) (B) | Fucosterol | Antiaging, Metalloproteinase inhibitors | [204] |
| 41 | *Ecklonia cava* (B) | Eckol, dieckol | Skin whitener | [204] |
| 42 | *Ishige foliacea* (B) | Phlorotannin | Downregulation of tyrosinase and melanin synthesis | [205,206] |
| 43 | *Laminaria ochroleuca* (B) | Polyphenol | Antioxidant | [207] |
| 44 | *Macrocystis pyrifera* (B) | Phlorotannin | Antioxidant, ROS scavenger | [208] |
| 45 | *Saccharina latissima* (B) | Phenol | Antioxidant | [209] |
| 46 | *Sargassum serratifolium* (B) | Sargachromenol | Anti-melanogenic | [210] |
| 47 | *Schizymenia dubyi* (R) | Phenol | Anti-melanogenic, tyrosinase inhibition | [210] |
| 48 | *Sargassum thunbergia* (B) | Thunbergol | Antioxidant | [211] |
| 49 | *Pyropia columbina* (R) | Phenol | Antioxidant | [212] |
| 50 | *Rhodomela confervoides* (R) | Bromophenol | Antioxidant | [213] |
| 51 | *Ulva prolifera* (G) | Phenol, flavonoid | Antioxidant | [214] |
| 52 | *Ulva rigida* (G) | Phenol | Antioxidant | [215] |
| 53 | *Ecklonia cava* (B) | Dioxinodehydroeckol | UVB protector | [216] |
| 54 | *Eisenia bicyclis, Ecklonia cava* subsp. *stolonifera* (as *E. stolonifera*) (B) | Ecokol | Anti-inflammatory, Tyrosinase inhibition | [217–219] |
| 55 | *Ecklonia cava* subsp. *stolonifera* (as *E. stolonifera*) (B) | Fucofuroeckol-A | UVB protector | [220] |
| 56 | *Cystoseira compressa* (B) | Fuhalol | Antioxidant | [221] |
| 57 | *Fucus vesiculosus* (B) | Fucophloroethol | Antioxidant | [222] |
| 58 | *Ecklonia cava* (B) | Eckstolonol | Antioxidant | [223] |
| 59 | *Ishige foliacea* (B) | Octaphlorethol-A | Antioxidant effects | [224] |
| 60 | *Eisenia bicyclis, Ecklonia cava* subsp. *stolonifera* (as *E. stolonifera*) (B) | Phlorofucofuroeckol-A | Hepatoprotective against oxidative stress, Tyrosinase inhibition | [225,226] |
| 61 | *Ecklonia cava* (B) | 2-phloroeckol, 2-O-(2,4,6-Trihydroxyphenyl)-6,6′-bieckol | Tyrosinase inhibition | [227] |
| 62 | *Ascophyllum nodosum, Fucus serratus, Himanthalia elongata, Halidrys siliquosa,* (B) | Phlorotannins | Antioxidant, Photoprotective | [228–230] |
| 63 | *Ecklonia cava* subsp. *stolonifera* (as *E. stolonifera*) (B) | Dioxinodehydroeckol | Downregulation of melanogenic enzymes that are namely TYR, TRP1, and TRP2 | [231] |

(B: Brown algae; G: Green algae; R: Red algae).

## 3. Extraction of Phenolic Compounds

There are several extraction techniques available for obtaining phenolic compounds; two general techniques are found: conventional and nonconventional extraction techniques. The conventional techniques include simple solid solvent extraction, whereas nontraditional techniques include microwave-assisted extraction, subcritical $CO_2$ extraction, ultrasound-assisted extraction, and pressurized liquid extraction, among others. The extraction and characterization of phenolic compounds from marine algae reported an interesting result as in Figure 5 [44–48].

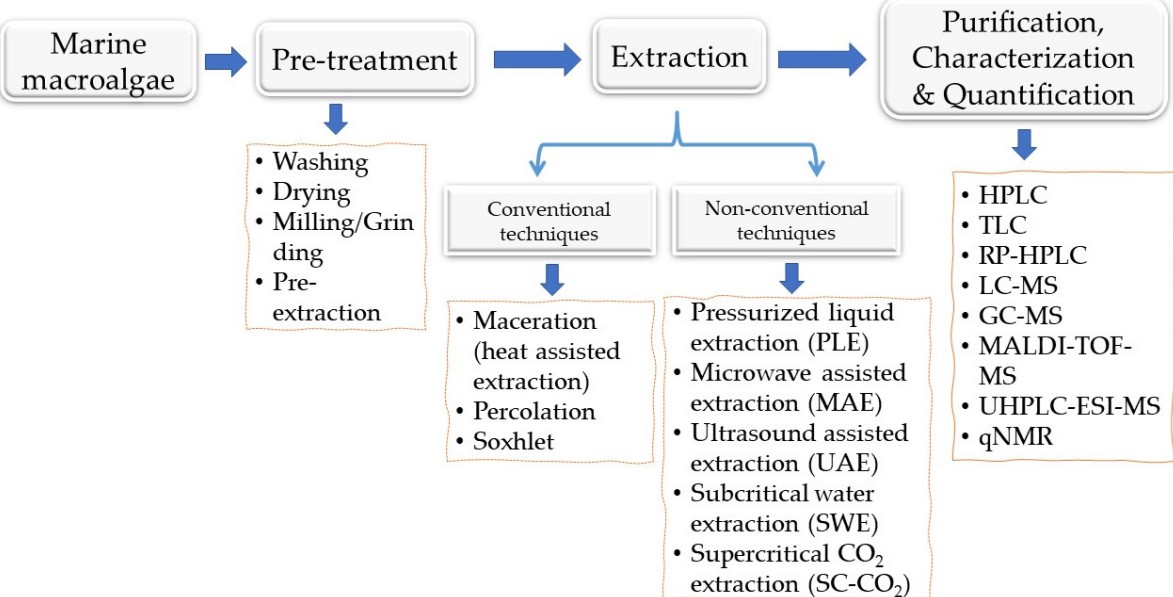

**Figure 5.** Techniques of extraction and characterization for phenolic compounds. Source: Santana-Gálvez and Jacobo-Velázquez [232].

The most important step is to select an appropriate extraction method, since many procedures of extraction are available nowadays. Traditional methods include heat-assisted extraction or maceration, percolation, and Soxhlet extraction as reported by Aires [233]. One of the classical methods is maceration, in which the components are extracted by submerging marine algae in an appropriate solvent or solvent combinations [234]. On a large scale, at the industrial level, ethanol is preferred as a solvent for extraction because of its economic benefit [235]. This procedure is widely applicable in current practice. In this method, methanol, ethanol, acetone, water, and ethyl ethanoate in different proportions are commonly utilized for extraction. The selection can be done based on polarity. Due to the hydrophilic nature of these compounds, hydroalcoholic solvent is the most effective for this process. Some previous studies have mentioned the combination of solvents, with acids such as citric acid, tartaric acid, or HCl potentially improving the extraction of phenolic compounds [236,237]. In traditional procedures, Soxhlet extraction provides better results of extraction in terms of yield, although this technique also presents some demerits such as the degradation of temperature-sensitive compounds as some phenolic acids, tannins, and anthocyanins require a large number of solvents and are time-consuming. Besides, this classical Soxhlet extraction method is a continuous process; the solvent can easily be recycled, and less time and less solvent are used than in maceration and percolation [238]. Moreover, the used extract of the selected algae is constantly being heated at the boiling point of the solvent and it may damage some temperature-sensitive components, which may affect further analysis [239]. Santos-Buelga et al. [236] reported the use of the Soxhlet method in the extraction of lipophilic compounds. Moreover, heat-assisted extraction can be divided into two steps. The first step is the faster step and the second one is slower. The faster method consists of a compound's transference from the matrix surface to the solvent whereas the slower method consists of diffusion from the matrix to the solvent. Extraction is mainly affected by the type of sample, type of solvent, temperature value, and time. The main disadvantage is that it requires filtration, decantation, or clarification to separate the solid parts when the extraction is done. It also requires a large number of solvents and takes a long time. Hence, these classical methods are not efficient and environmentally friendly due to the high requirements of the organic solvents [240]. With advancements, new techniques have evolved to improve the efficacy and accuracy of extraction.

A non-conventional technique, Pressurized Liquid Extraction (PLE), also known as extraction with pressurized solvent, includes high pressure (1 to 15 MPa), short processing time, and temperature ranges of about 50 to 200 °C using a low volume of nontoxic solvent and thus being considered a green technology. Otero et al. [241] observed a highest extraction yield of 37% for 80 °C and 52% for 160 °C using diluted ethanol from the brown alga *Laminaria ochroleuca* (Phaeophyceae) at 100 bars. Microwave-assisted extraction is mainly used for the extraction of polyphenols and polysaccharides. This method can be performed in open (at atmospheric pressure) or closed (higher than atmospheric pressure) vessels. In this method, electromagnetic waves cause changes in cell structures. Two mechanisms, ionic conduction, and dipole rotation, transform electromagnetic energy into calorific energy [242,243]. There are many affecting factors such as the type of extraction, frequency of microwave, solid-to-solvent ratio, temperature, pressure, and time. Besides, the demerit is that high microwave power and elevated temperature may degrade phenolic compounds [244]. Cikoš et al. [243] showed the merits of ultrasound-assisted extraction (UAE) for the extraction of phenolic compounds from algae including low temperature, short times, and low amounts of solvent. However, the ultrasonication time can increase the temperature, which may affect the stability of the phenolic compounds. Besides, there are some other affecting factors such as frequency, power, time, temperature, and solid: solvent ratio [245]. This method applies ultrasound waves with a frequency between 20 kHz and 100 kHz, which creates bubbles due to pressure differences. Then, the bubbles collapse and cavitation occurs, causing the near liquid–solid interface breakdown of particles with the release of bioactive compounds to the matrix. The subcritical water extraction (SWE) method requires an application of water at a higher temperature (100–374 °C) than its boiling point under high pressure (10–60 bar) to maintain its liquid state for 5–10 min. In this technique, pressure, time, temperature, and the selection of the solvent are affecting factors for extraction [246,247]. In the SC-$CO_2$ method, $CO_2$ is a nontoxic gas used as a supercritical fluid, so the fluid behaves like a liquid simultaneously, which makes extraction easier. Ethanol can be used to change the polarity of $CO_2$, while extraction, low temperature, and low pressure are used to degrade the phenolic compounds. Moreover, varieties of seaweed species were studied and explored for a great variety of biochemicals and their skin benefits.

## 4. Commercial Availability of Seaweed-Based Cosmetic Products

Marine algae have received more attention recently in cosmetics. Several skin cosmetic products are available in the market, some of them prepared by using algal extract, whereas some contain extracted bioactive compounds from potential marine algae. Nowadays, several cosmetic companies are using marine algae extracts and compounds in cosmetic preparations, as an active ingredient or as an excipient, gelling, thickening, preservative, additive, aroma, or fragrance agent [248]. For example, red alga *Gracilaria* sp. extracts are integrated into different products, such as A-Gel, Sealaria (Kfar Hess, Israel; https://www.sealaria.com/our-products/, accessed on 23 September 2022), facial masks by Balinique (Miami, FL, USA; https://www.gsg-creative.com/cases/balinique, accessed on 23 September 2022), and hydrating creams by Thalasso (Rosa Graf, Stamford, USA; https://skincare.rosagraf.com/product-category/thalasso/, accessed on 23 September 2022) [249]. Helioguard® 365 (Mibelle Biochemistry; https://mibellebiochemistry.com/helioguardtm-365, accessed on 21 September 2022) is a cosmetic ingredient complex that was formulated by using mycosporine-like amino acids derived from the red alga *Porphyra umbilicalis* (Rhodophyta), which has a powerful UV-protective capacity. This product proved suitable to use daily and boasts photoaging benefits. Besides, a product prepared by OSEA Malibu (Los Angeles, USA; https://oseamalibu.com/products/undaria-algae-oil, accessed on 24 September 2022), *Undaria* algae body oil, contains *Undaria pinnatifida* powder, which confers antioxidant benefits and improves skin nourishment and firmness. "Hyaluronic Sea Serum" is prepared by mixing *Codium fragile* (Chlorophyta) extract with other natural extracts and ingredients to improve hydration, minimize lines, and enhance firmness (shorturl.at/lq148).

There are some other products such as W2 SPF 50 PA+++ Red Seaweed (Life Essentials Personal Care Pvt Ltd., Haryana, India; shorturl.at/lrS28), which contains red alga extract that prevents dehydration of the skin and provides photoprotection benefits; Seaweed Cleansing Soap (Mario Badescu Skin Care Inc., New York, NY 10022 https://www.mariobadescu.com/product/seaweed-cleansing-soap, accessed on 24 September 2022), which contains seaweed grains and Bladderwrack (Seaweed) extract that contribute to nourishment, creamy cleansing, and soothing and gentle mineral exfoliation benefits; Seaweed Oil Control Gel Cream (The Body Shop International Limited, West Sussex, UK; https://www.thebodyshop.com/en-gb/face/moisturisers/seaweed-oil-control-gel-cream/p/p000181, accessed on 23 September 2022), which controls shine, hydration, skin protection, and nourishment; Sea Algae Daily Repair Serum (Prolixr, India; https://prolixr.in/products/sea-algae-daily-repair-serum, accessed on 23 September 2022), which replenishes moisturization and improves skin elasticity; and in addition other products, Seaweed Oil-Control Gel Cream (The Body Shop International Limited, UK), prepared by using the extract of the brown alga *Fucus vesiculosus* with other natural ingredients that help to maintain oil balance and excess sebum for a matte and shine-free complexion. Likewise, Sea Algae Daily Repair Ace Serum (FURR, Pee Safe, India; https://furr.in/products/furr-daily-repair-face-serum, accessed on 18 October 2022) strengthens the elastin tissues, revitalizes the skin and reduces shine. Another market-available product is Universal Face Oil by MARA Beauty (Queenstown, New Zealand; https://themarabeauty.com/products/algae-moringa-universal-face-oil, accessed on 18 October 2022) that is made by mixing algae plus moringa. In this product, algae play a proprietary role that enhances the natural hyaluronic acid synthesis and is loaded with phytonutrients and fatty acids to improve the plumpness, firmness, and smoothness of the skin. Moreover, Green Confertii Extract-NS (Gyeonggi-do, Republic of Korea; https://cosmetics.specialchem.com/product/i-the-garden-of-naturalsolution-green-confertii-extract-ns, accessed on 18 October 2022) contains an extract of *Ulva compressa* (formerly *Enteromorpha compressa*) (Chlorophyta), which is rich in bioactive compounds, polysaccharides, flavonoids, tannins, and acrylic acid. This extract possesses antioxidant, antiallergic, and antimicrobial activity.

## 5. Conclusions

The macroalgae-derived phenolic compounds are scarce and further exploration will create a good library of bioactive chemicals and enhance the possibility of discovering new compounds in different types of skin cosmetic preparations. Hence, phycological research, mainly isolation, extraction, and the characterization of seaweed species, will improve the cosmetic market commercially. The main focus is on the concentration of bioactive compounds present in macroalgal species, which creates a real problem at the formulation level. Polyphenolic compounds and other classes of chemical compounds are the attention of ongoing research. Their study is very limited and there is a lack of clarity about the in vivo effects of seaweed-derived phenolic compounds and their interaction with human cells. These problems can be overcome by using various methodologies and determination methods that evaluate at a deeper level to make them safer. More studies and research are needed on the characterization of phytochemicals, extraction, characterization, and in vitro and in vivo study for toxicity by diversified methodology. Overall, new research studies are required to analyze and fully understand their biological benefits in cosmetic formulation and on the skin to make the cosmetic sector sustainable.

Ultimately, many more seaweed species will require study and characterization for their application in cosmetic formulation and to understand their skin benefits. Due to their tremendous number of applications and various biological benefits, marine macroalgae are gaining attention and becoming increasingly attractive in the exploration of the skin cosmetic properties of their natural bioactive extracts and formulations.

**Author Contributions:** Conception and design of the idea: H.S.K. and L.P.; Design of tables and figures: H.S.K. and L.P.; Writing and bibliographical research: H.S.K. and L.P.; Supervision and manuscript revision: H.S.K. and L.P. All authors have read and agreed to the published version of the manuscript.

**Funding:** This work was financed by national funds through the FCT—Foundation for Science and Technology, I.P., within the scope of the project LA/P/0069/2020 granted to the Associate Laboratory ARNET, UIDB/04292/2020 granted to MARE—Marine and Environmental Sciences Centre.

**Institutional Review Board Statement:** Not applicable.

**Informed Consent Statement:** Not applicable.

**Data Availability Statement:** Not applicable.

**Conflicts of Interest:** The authors declare no conflict of interest.

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
