# Peer review of "Dermo-Cosmetic Benefits of Marine Macroalgae-Derived Phenolic Compounds"

_applsci, doi:10.3390/app122311954_

Round 1

Reviewer 1 Report

The review is written nicely. It can be accepted in its present form.

Author Response

Reviewer 1

The review is written nicely. It can be accepted in its present form.

Response: Thank you very much

Reviewer 2 Report

The manuscript requires a thorough review of its spelling and grammar

Line 71. Are they really economic?

Line 77. Please write correctly the scientific names of algae

  phycochemicals??? It is right?

Line 123. What enzymes does it inhibit? A couple of examples would be good.

Regarding the figures, I don't know if only the source should be mentioned or if permission should be obtained from the publisher that published such a figure for its use. please check that

Line 224-226. Please restructure this sentence. The idea is not well understood

Line 258-268. In this paragraph, where does it talk about cosmetic effects?

It is recommended in the table to only talk about the cosmetic benefits, since it is the objective of your review.

I do not know how acceptable it is to use the trade names of products that have been made with compounds from algae

Is the discussion section necessary? If it is, it's really short to be considered a discussion

Author Response

Reviewer 2

All corrections are yellow highlighted.

Line 71. Are they really economic?

Response: Changed it. Lines: 70-71

Line 77. Please write correctly the scientific names of algae

Response: Lines: 77-78, 102

phycochemicals??? It is right?

Response: I wrote it in a meaning of seaweed (study about seaweeds: Phycology) based bioactive compounds. But i replaced in Phytochemicals instead of Phycochemicals. Lines: 13, 106, 491

Line 123. What enzymes does it inhibit? A couple of examples would be good.

Response: Lines: 118-123, also cited references in the list

Regarding the figures, I don't know if only the source should be mentioned or if permission should be obtained from the publisher that published such a figure for its use. please check that

Response: Permission obtained

Line 224-226. Please restructure this sentence. The idea is not well understood

Response: reformulated, Lines: 260-266

Line 258-268. In this paragraph, where does it talk about cosmetic effects?

Response: Reformulated and references also updated Lines: 297-320

It is recommended in the table to only talk about the cosmetic benefits, since it is the objective of your review.

Response: By considering reviewer’s suggestion, removed unnecessary, cosmetic unrelated benefits from the table 1, renumbered the serial numbers as well as reference list improved.

I do not know how acceptable it is to use the trade names of products that have been made with compounds from algae

Response: Lines 437-475, The reason behind using the trade names of products is to show the commercially available seaweed based products.

Is the discussion section necessary? If it is, it's really short to be considered a discussion

Response: Not necessary, by considering your suggestion as well as suggestion of reviewer 3, we removed discussion part.

Reviewer 3 Report

in line 78 convert class to  phylum

rewrite lines 107-110.

line 163 converts Chemical to chemical

line 130 converts title 2. to characterization and types of phenolic compounds

Line 242 converts Bromphenol to bromophenol

in part types of phenolics add more details about all types of phenolics 

line 302 adds a figure to illustrate the extraction of phenolics

why did the authors add part of the discussion if you add it you must add materials and results

the references are very old please updated it 

Author Response

Reviewer  3

Green highlighted

in line 78 convert class to phylum

Response: Incorporated

rewrite lines 107-110.

Response: rewrite, lines 106-107

line 163 converts Chemical to chemical

Response: Line 200

line 130 converts title 2. to characterization and types of phenolic compounds

Response: Rewrite

Line 242 converts Bromphenol to bromophenol

Response: Rewrite, Line 282

in part types of phenolics add more details about all types of phenolics 

Response: Improved and rewrite Lines: 130-168, 260-266, 303-309, 311-320.

line 302 adds a figure to illustrate the extraction of phenolics

Response: Added, line 362

why did the authors add part of the discussion if you add it you must add materials and results

Response: I removed discussion part from the ms and improved other parts of the text.

the references are very old please updated it 

Response: Updated, References highlighted in green.

Reviewer 4 Report

Kalasariya and Pereira have revised the literature on the cosmeceutical effect of marine macroalgae-derived phenolic compounds.

The topic is interesting and current. However, the whole manuscript needs extensive rewriting. The MS articulate very poorly. There is no consistency and flow in the reading.

For example, section 2 starts with role of phenolic compounds in skin benefits, then come phytochemicals and extraction

Also, many awkward sentences have been used throughout the manuscript starting from the abstract to the conclusion

The whole manuscript needs extensive English language editing and sentences need to write in standard English

Line 13: “phyco-chemical” should be “phytochemicals” Please follow throughout the text.

Line79-82: awkward sentence

Line 108-119: too general sentences, please remove or improve.

Author Response

Reviewer 4

All corrections are highlighted Turquoise blue

Line 13: “phyco-chemical” should be “phytochemicals” Please follow throughout the text.

Response: Followed throughout the text, Lines: 13, 106, 491

Line79-82: awkward sentence

Response: Lines 79-82, turquoise blue highlight

Line 108-119: too general sentences, please remove or improve.

Response: We removed some general sentence and reformulated some sentences. Lines: 107-115

Round 2

Reviewer 2 Report

The authors have responded to each of the observations

Reviewer 3 Report

The authors improved the review and do all corrections 

Reviewer 4 Report

Accepted